

# Proteomic study on nintedanib in gastric cancer cells

Xiaohua Dong[1,2,3], Liuli Wang[1], Da Wang[2], Miao Yu[2,3], Xiao jun Yang[1,2,3] and Hui Cai[1,2,3]

[1] The First School of Clinical Medicine, Lanzhou University, LanZhou, China
[2] Department of General Surgery, Gansu Provincial Hospital, LanZhou, China
[3] Key Laboratory of Molecular Diagnostics and Precision Medicine for Surgical Oncology in Gansu Province and NHC Key Laboratory of Diagnosis and Therapy of Gastrointestinal Tumor, Gansu Provincial Hospital, LanZhou, China

## ABSTRACT

**Background**. Gastric cancer is a very common gastrointestinal tumor with a high mortality rate. Nintedanib has been shown to significantly reduce tumor cell proliferation and increase apoptosis in gastric cancer cells *in vitro*. However, its systemic action mechanism on gastric cancer cells remains unclear. A high-throughput proteomic approach should help identify the potential mechanisms and targets of nintedanib on gastric cancer cells.

**Methods**. The effects of nintedanib on the biological behavior of gastric cancer cells were evaluated. A cytotoxic proliferation assay was performed to estimate the half maximal inhibitory concentration ($IC_{50}$). AGS cells were divided into control, and nintedanib-treated groups (5 $\mu$M, 48 h), and differential protein expression was investigated using tandem mass tags (TMT) proteomics. The molecular mechanisms of these differentially expressed proteins and their network interactions were then analyzed using bioinformatics, and potential nintedanib targets were identified.

**Results**. This study identified 845 differentially expressed proteins in the nintedanib-treated group (compared to the control group), comprising 526 up-regulated and 319 down-regulated proteins. Bioinformatics analysis revealed that the differentially expressed proteins were primarily enriched in biological pathways for branched-chain amino acid metabolism, steroid biosynthesis, propionate metabolism, fatty acid metabolism, lysosome, peroxisome, and ferroptosis. Key driver analysis revealed that proteins, such as enoyl-CoA hydratase and 3-hydroxyacyl CoA dehydrogenase (EHHADH), isocitrate dehydrogenase 1 (IDH1), acyl-CoA oxidase 1 (ACOX1), acyl-CoA oxidase 2 (ACOX2), acyl-CoA oxidase 3 (ACOX3), and acetyl-CoA acyltransferase 1 (ACAA1) could be linked with nintedanib action.

**Conclusion**. Nintedanib inhibits the proliferation, invasion, and metastasis of gastric cancer cells. The crossover pathways and protein networks predicted by proteomics should provide more detailed molecular information enabling the use of nintedanib against gastric cancer.

Corresponding author
Hui Cai, caialonteam@163.com

## BACKGROUND

Gastric cancer is one of the most common gastrointestinal tumors worldwide. Based on the 2020 global cancer statistics, gastric cancer ranks fifth in incidence and fourth in mortality among all cancer types (*Sung et al., 2021*). Because of its insidious onset and the lack of effective early screening, most gastric cancer patients are diagnosed at a comparatively late stage. Although various treatments such as surgery, chemotherapy, radiotherapy, and immunotherapy have been used, the 5-year survival rate of gastric cancer patients has not significantly improved (*Cheng et al., 2023*). Therefore, the development of new therapeutic drugs for gastric cancer remains a hot topic in the cancer field, even though drug development is a time-consuming and costly process. One way to avoid these problems is to reuse existing drugs with cytotoxicity. Drug reuse has been employed in cancer treatment for years, generally providing good results (*Zhang et al., 2020*).

Nintedanib is a novel triple receptor tyrosine kinase inhibitor. It blocks the vascular regeneration and fibrosis pathways controlled by vascular endothelial growth factor receptor (VEGFR), fibroblast growth factor receptor (FGFR), proto-oncogene tyrosine-protein kinase Src, and Fms-like tyrosine kinase 3 (FLT-3) (*Hilberg et al., 2008*). Nintedanib is clinically approved to treat idiopathic pulmonary fibrosis, and is the first targeted therapy. Previous studies have provided evidence that nintedanib is also effective in tumor treatment. Clinical trials have demonstrated that nintedanib combined with docetaxel is safe and effective in treating advanced lung adenocarcinoma (*Corral et al., 2019*). Nintedanib targets PAX5 gene fusion in children with acute lymphoblastic leukemia (*Fazio et al., 2022*). It also exerts an excellent therapeutic effect on metastatic esophageal gastric cancer mediated by multiple oncogenic receptor tyrosine kinase (RTK) gene amplification (*Won et al., 2019*). Preclinical experiments revealed that nintedanib alone or combined with other chemotherapeutic drugs significantly reduces tumor cell proliferation and increases apoptosis in gastric cancer cells *in vitro* (*Awasthi et al., 2018*). However, the specific mechanism of nintedanib in gastric cancer cells remains unclear, requiring further clarification.

Proteomics is a study of the composition and variation of proteins in organisms, using the proteome as the object. Importantly, proteomics can be used to explore drug action mechanisms in cells and tissues (*Nojima et al., 2023*; *Salovska et al., 2023*; *Wang et al., 2022*). Indeed, proteomics has already been used to provide an insight into the molecular mechanism of nintedanib. *Falcomatà et al. (2022)* used proteomics to determine the target and tumor suppressor pathway of nintedanib on KRAS mutant pancreatic ductal adenocarcinoma (PDAC). In addition, proteomics was used to reveal the protein molecular characteristics of the bleomycin-induced pulmonary fibrosis mouse model after nintedanib treatment (*Principi et al., 2023*). *Landi et al. (2020)* also applied functional proteomics to analyze the serum samples of idiopathic pulmonary fibrosis (IPF) patients before and after one year of nintedanib treatment, with the results indicating molecular pathway biomarkers of drug-induced therapeutic responses. In the current study, we have applied quantitative proteomics labeling to explore the protein targets and molecular mechanisms of nintedanib within gastric cancer cell lines.

# MATERIALS AND METHODS

## Cell lines and cell culture

Procell Life Science & Technology Co., Ltd. supplied the gastric cancer cell line AGS (CL-0022). The human gastric adenocarcinoma cell line MKN28 was purchased from Shanghai F&H Biotechnology Co. The cell lines were cultured in RPMI-1640 medium (# C11875500BT; Gibco, Carlsbad, CA, USA) supplemented with 10% fetal bovine serum (# 10099141; Gibco, Carlsbad, CA, USA) and 1% penicillin-streptomycin (#15140122; Gibco, Carlsbad, CA, USA) and incubated at 37 °C with 5% $CO_2$.

## Drugs and reagents

Nintedanib was procured from TargetMol at a purity >99.92%. Nintedanib was dissolved in dimethyl sulfoxide (#D8372; Solarbio, Beijing, China) at 10 mM concentration and stored at −20 °C. It was diluted in a culture medium before use.

## Cell viability assay

A CCK-8 assay was used to evaluate the effect of nintedanib on cell viability. First, AGS and MKN28 cells were seeded at 4,000 cells/ well and 6,000 cells/ well, respectively, and the plates were then incubated at 37 °C for 24 h under 5% $CO_2$. Next, these cells were treated with different concentrations of nintedanib (0 μM, 0.01 μM, 0.1 μM, 1 μM, 10 μM, and 100 μM). Each nintedanib dose was added to five parallel wells ($n = 5$). Cell viability was then evaluated at 24 h, 48 h, and 72 h using a CCK-8 kit. For the assay, 10 μL CCK-8 reagent (#K1018-5; APExBIO, Boston, MA, USA) was added to each well, and the plate was incubated for 1 h. The OD value of each well was then measured at 450 nm using a Multiskan FC (Thermo Fisher Scientific, Waltham, MA, USA). All results are expressed as percentage cell viability.

## Cell scratch assay

AGS and MKN28 cells ($5 \times 10^5$ cells/well) were seeded in 6-well plates and incubated until achieving almost 100% confluence. Next, scratching was performed using a 200 μL plastic pipette tip. The plates were then cultured with different nintedanib concentrations (0 μM, 1 μM, 5 μM, 10 μM) in a serum-free medium. Wound closure was measured at 0 h and 48 h after scratching. The experiment was performed in triplicate.

## Transwell assay

A Transwell chamber (Millipore, Burlington, MA, USA) was used to assess cell invasion. Cells were added at a density of $1 \times 10^5$/ well to the upper chamber containing medium with nintedanib (final concentration, 0 μM, 1 μM, 5 μM, or 10 μM ). The lower chamber was filled with medium containing 20% fetal bovine serum. After incubating the chambers for 48 h, cells that did not pass through the membrane were wiped using a cotton swab. The cells at the bottom of the chamber were fixed with 4% paraformaldehyde (30 min), stained with 0.1% crystal violet (20 min), and observed under an inverted microscope. The number of cells that migrated across the membrane was counted in five random fields.
## 3D cell sphere culture

The ability of cells to form a 3D cell sphere was evaluated using CellCarrier-96 Ultra Microplates (6055308; Perkin Elmer, Shelton, CT, USA). First, cells were centrifuged, resuspended, and added to Ultra Microplates at a concentration of $8 \times 10^3$ cells per well. Nintedanib was then added to each well (at 0 µM , 1 µM , 5 µM , and 10 µM concentrations), and the microplates were incubated for 48 h. The resulting sphere shapes were then observed using a 4 × phase contrast microscope. ImageJ was used to measure and analyze the area of the spheroids.

## Proteomics

AGS cells were seeded in a 100 mm culture dish (CORNING), and incubated for 24 h. Next, nintedanib (5 µM) was added to the cells, and the dishes were returned to the incubator. After 48 h, the gastric cancer cells in the dishes were scraped and rapidly frozen in liquid nitrogen. These cells were then processed for subsequent analysis. A tandem mass tags (TMT) proteomic analysis was performed with the help of Oyi Biotechnology Co., LTD (Shanghai, China). First, dithiothreitol (DTT) was added to each of the protein samples to a final concentration of 10 mM. The samples were then incubated at 55 °C for 30 min, and cooled to room temperature. Iodoacetamide solution (final concentration, 10 mM) was added, and the cells were incubated for 15 min at room temperature. To precipitate total protein in the samples, acetone was added, and the samples were incubated at −20 °C for 4 h. After re-solubilization of the precipitate in TEAB (200 mM), trypsin was added at 1/50th of the sample mass, and the samples were incubated at 37 °C overnight. TMT-10 reagents were used for peptide labeling.

LC-MS/MS analysis was performed using the Easy-nLC 1200 liquid chromatography system coupled to a Q Exactive HF mass spectrometer (both from Thermo Fisher, Waltham, MA, USA). The peptides were first captured on an Agilent Zorbax Extend-C18 column (2.1 × 150 mm, 5 µM ), and then separated on an Acclaim PepMap RSLC column (75 µM × 50 cm, RP-C18, Thermo Fisher) at a flow rate of 300 nL/min for 60 min. Before MS/MS analysis, full MS survey scans were implemented (scan range, 350–1,500 m/z; resolution, 60,000), and the top 20 most abundant peaks were selected for MS/MS analysis. MS/MS spectra were recorded in a data-dependent mode, and a normalized higher energy collision dissociation of 32 was used for fragmentation. All MS/MS scans were performed at a resolution of 45,000. The AGC target of MS/MS was set to 2e5 with a maximum injection time of 70 ms and a dynamic exclusion of 30s.

Next, the raw data were analyzed using ProteomeDiscoverer 2.4.1.15 (ThermoFisher Scientific, Waltham, MA, USA) for protein identification. The screening criteria for differentially expressed proteins were: Score Sequest HT >0; unique peptide ≥ 1; and false discovery rate (FDR) <0.01. Kyoto Encyclopedia of Genes and Genomes (KEGG) and gene ontology (GO) enrichment analyses were used to describe the class and function of identified proteins. In addition, an interaction network analysis of significantly and differentially expressed proteins was performed using the STRING database. Mergeomics2.0 network tool was used for key driver analysis (KDA) to identify the critical regulators of relevant pathways and networks.

## Statistical analysis

Statistical analysis was performed using GraphPad Prism version 9.0. The experiments were conducted in technical triplicates. All the data are presented as mean ± standard deviation (SD) of triplicate experiments. A Student's $t$-test was used to compare the means of two groups, and a one-way ANOVA was used to compare the means of multiple groups. A value of $p < 0.05$ was used to determine statistical significance.

# RESULTS

## Nintedanib inhibits proliferation, invasion, and metastasis in gastric cancer cells

The effects of nintedanib were evaluated based on the biological behavior of gastric cancer cells. The tests were performed with MKN28, a highly differentiated gastric cancer cell line, and with AGS, a poorly differentiated gastric cancer cell line. The CCK-8 assay results revealed that nintedanib treatment inhibited gastric cancer cell viability in a dose-dependent and time-dependent manner. After 48 h exposure, nintedanib exhibited an $IC_{50}$ value of $5.3 \pm 0.9$ µM with AGS cells, and an $IC_{50}$ value of $6.1 \pm 1.2$ µM with MKN28 cells (Fig. 1A). Next, a transwell migration assay was used to assess the effect of nintedanib on cell invasion. The numbers of migrated AGS and migrated MKN28 cells were significantly higher without treatment, compared with cells treated (for 48 h) with 1 µM, 5 µM , and 10 µM nintedanib (Fig. 1B). The cell scratch assay demonstrated that nintedanib significantly inhibited the migration of AGS and MKN28 cells. In contrast, the inhibitory effect of 1 µM was not statistically significant in MKN28 cells (Fig. 1C). Finally, a 3D cell sphere-forming assay was utilized to mimic the proliferation pattern of cells *in vivo*. The results indicate that nintedanib inhibited gastric cancer cell proliferation dose-dependently at 1 µM, 5 µM, and 10 µM (Fig. 1D).

## Identification of differentially expressed proteins

A TMT-labeled quantitative proteomic analysis was implemented to examine the alterations in protein expression and in biological pathways in gastric cancer cells caused by nintedanib. Four biological replicate experiments were performed in the control and nintedanib-treated groups, and around 7,823 proteins and 81,592 peptides were identified. Differential proteins were defined as proteins with a fold change ≥ 2 or ≤ 1/2 and a $p$-value <0.05. A total of 845 significantly differentially expressed proteins were screened, 526 upregulated proteins and 319 down-regulated proteins (Fig. 2A). In addition, volcano plots and cluster heat maps are used to show significance distribution and fold change for the differentially expressed proteins in the drug-treated group (Figs. 2B–2D).

## GO enrichment analysis

Gene Ontology analysis is used to define the roles and behavioral functions of genes and proteins according to biological process, cellular component, and molecular function. GO analysis of differentially expressed proteins predominantly revealed the perturbation of biological processes responsible for fatty acid oxidation, cholesterol metabolic processes, response to toxic substances, translation initiation, and the formation

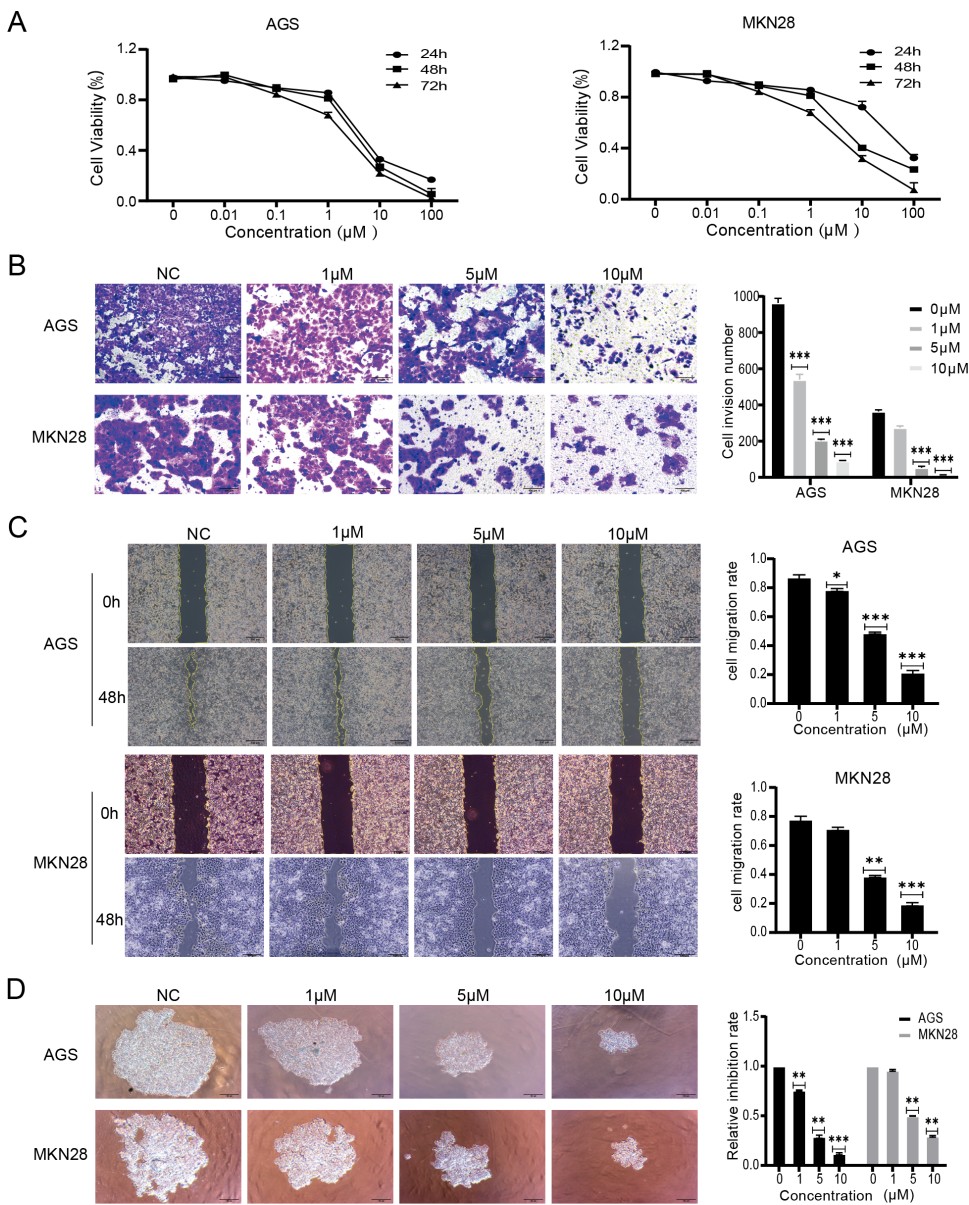

**Figure 1** **Nintedanib attenuates proliferation, migration, and invasion of gastric cancer cells.** (A) CCK-8 assays were used to assess the viability of AGS cells and MKN28 cells in the following groups (0 µM, 0.01 µM, 0.1 µM, 1 µM, 10 µM, and 100 µM nintedanib). OD values were obtained at 24 h, 48 h, and 72 h (B) Transwell assays were used to evaluate the invasive capacity of AGS cells and MKN28 cells (scale bar, 100 µm). (C) Cell scratch assays were employed to assess the impact of nintedanib on cell migration ability (scale bar, 100 µm). (D) 3D cell sphere-forming assays were conducted to simulate *in vivo* cell growth patterns, and to investigate the effect of nintedanib on cell proliferation ability (scale bar,100 µm). $^{*}p < 0.05$, $^{**}p < 0.01$, $^{***}p < 0.001$.

of cytoplasmic translation initiation complexes (Fig. 3A). In addition, GO analysis primarily established perturbation of cellular components in the cytoplasm, exosomes, membranes, mitochondria, and endoplasmic reticulum. Finally, GO analysis also provided especially
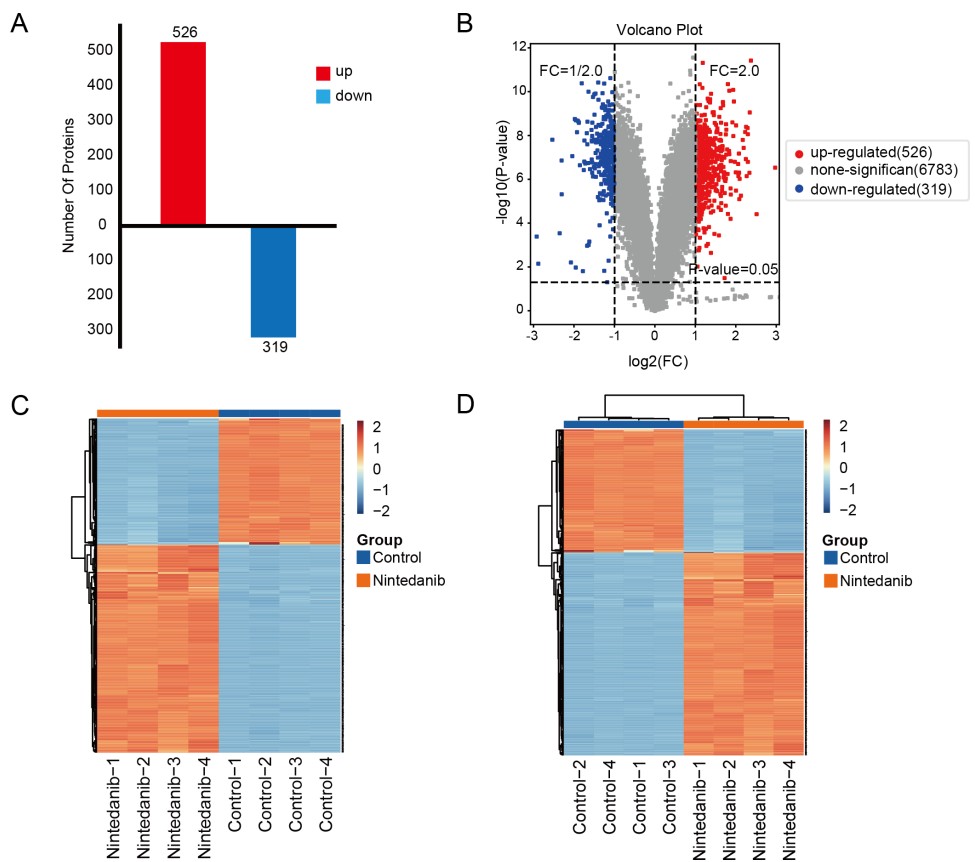

**Figure 2** **Results of differentially expressed proteins analysis.** (A) Bar graphs showing the numbers of up-regulated and down-regulated proteins. (B) Volcano plot of differentially expressed proteins. (C, D) Heat maps showing the proteins positively or negatively correlated with nintedanib treatment (C) sample clustering; (D) sample non-clustering.

firm evidence for the perturbation of molecular functions for identical protein binding, enzyme binding, calmodulin binding, translation initiation factor, and oxidoreductase activity. The top six GO entries selected in this study (using ListHits >3 and <50) sorted by -$\log_{10}$ $p$-value were in descending order: cytoplasmic translation initiation complex formation, eukaryotic 43S pre-initiation complex, fatty acid-oxidation, eukaryotic translation initiation factor 3 complex, and eukaryotic 48S pre-initiation complex (Fig. 3B).

## Pathway annotation

Differentially expressed proteins in the nintedanib-treated group were analyzed by stratified pathway analysis with the KEGG database (combined with KEGG annotation results). The significance of differentially expressed protein enrichment in each pathway entry was determined as a $p$-value from the hypergeometric distribution. The enriched pathways of the differentially expressed proteins were primarily associated with steroid biosynthesis, branched-chain amino acid metabolism, fatty acid degradation, peroxisome, PPAR signaling pathway, and ferroptosis (Fig. 4A). Finally, level 3 KEGG classification of the

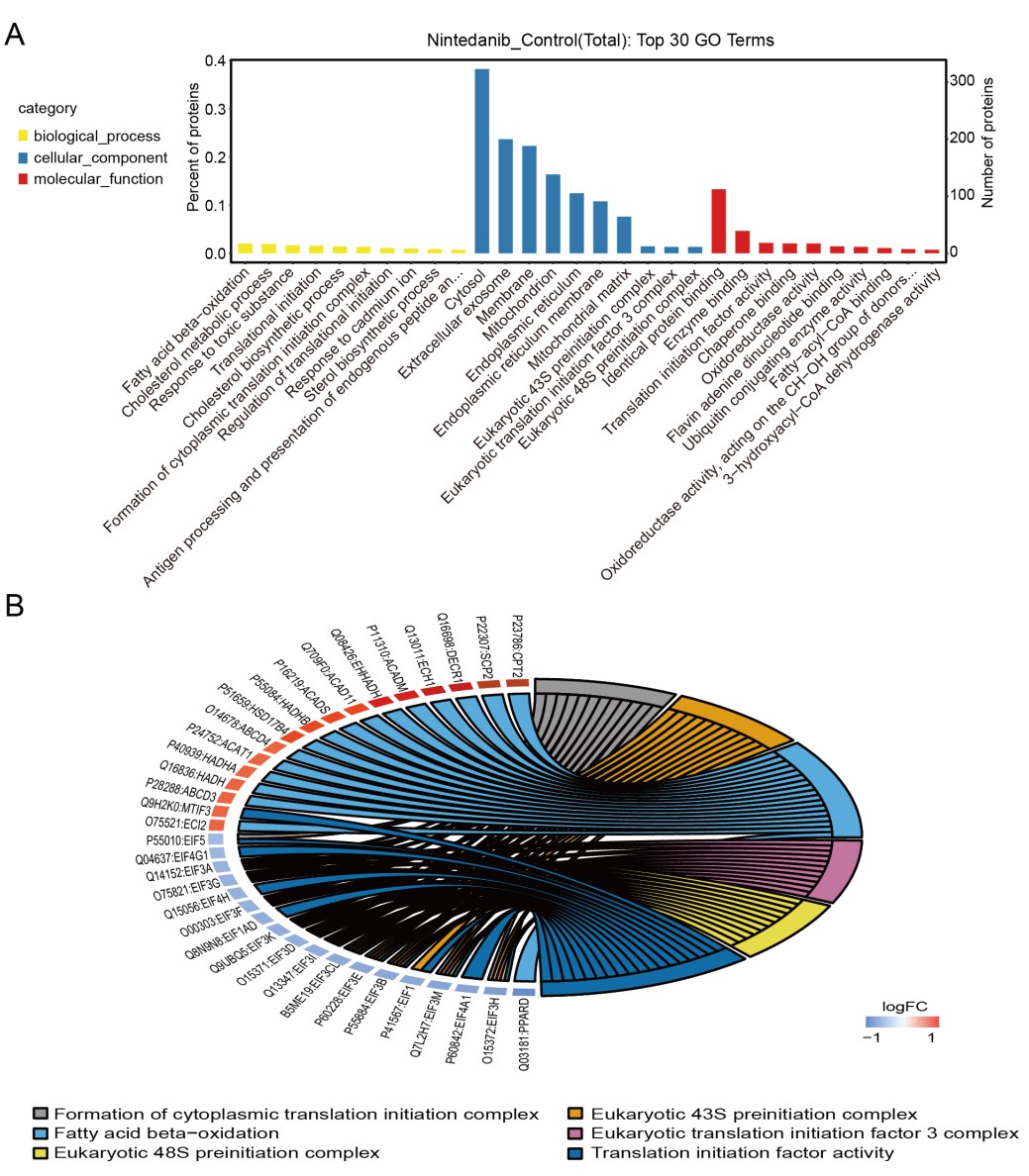

**Figure 3 GO annotation of differentially expressed proteins.** (A) Gene ontology enrichment results for differentially expressed proteins in the nintedanib group (compared with the control group). Yellow, biological processes; blue, cellular components; red, molecular functions. (B) GO enrichment analysis chord diagram of differentially expressed proteins.

differentially expressed proteins identified metabolic pathways involved in metabolism, cellular processes, genetic information processing, and organismal systems (Fig. 4B).

## GSEA enrichment analysis

GSEA analysis of sample data was performed to explore the potential molecular mechanisms of nintedanib on gastric cancer cells, while compensating for certain important gene information that could have been missed during GO/KEGG enrichment analysis. Our results are significantly improved in clusters associated with cell metabolism.

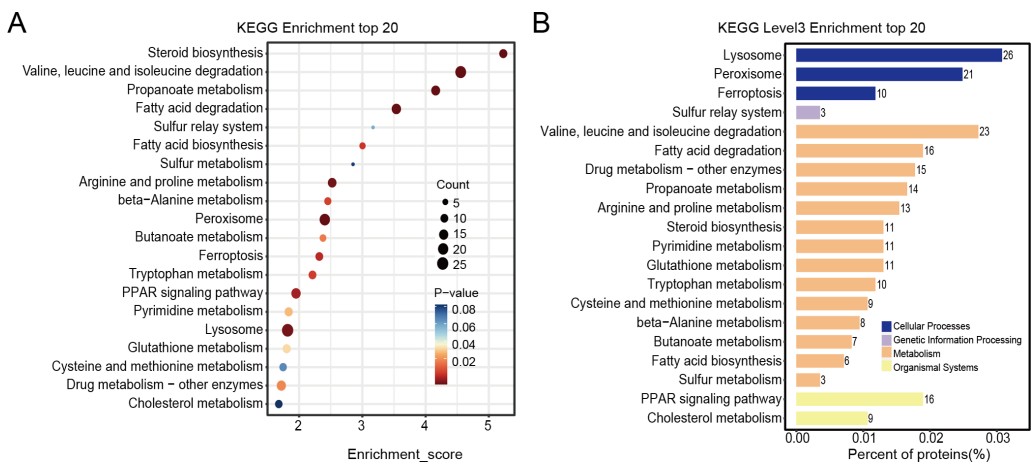

**Figure 4** **KEGG enrichment analysis results of differentially expressed proteins.** (A) Pathway enrichment results for differentially expressed proteins in the nintedanib group (compared with the control group). Color and count represent enrichment significance and number of differentially expressed proteins enriched in the pathway, respectively. (B) Distribution of differentially expressed proteins at KEGG Level 3 ($y$-axis denotes the name of each level 3 pathway).

The upregulated terms were "valine, leucine, and isoleucine degradation", "oxidative phosphorylation", "diabetic cardiomyopathy", "thermogenesis", "lysosome peroxisome", "propanoate metabolism", "fatty acid degradation", and "steroid biosynthesis" (Figs. 5A–5I).

## Protein–protein interaction analysis

The STRING database was used to construct a protein–protein interaction (PPI) network of differentially expressed proteins in the control and drug-treated groups. The differentially expressed proteins were analyzed in the STRING database by selecting the native/ proximate species (blast $e$-value: 1e−10) while retrieving the interactions of differential proteins, selecting the top 25 proteins for connectivity, and plotting the interactions network. The top 25 nodes in terms of node connectivity were visualized using the Python package 'network' and displayed with the protein gene name. The connected node size represents the connectivity level, with larger circles depicting higher connectivity. The hub proteins identified, CAT, PLEKHA4, HSP90AB1, TXN, EEF2, H6PD, and PDIA3, may play important roles in the treatment of gastric cancer using nintedanib (Fig. 6).

## Key driver analysis

A key driver analysis was performed independently depending on a list of selected genes and a given network as input. This analysis helps identify the interactions of disease genes and the presence of crucial hub nodes in the network regulating disease genes. Our analysis predicted potential targets in the network of molecules acting on gastric cancer cells treated with nintedanib from a consideration of our proteomic differential protein results. The top 15 genes identified are shown in Table 1.

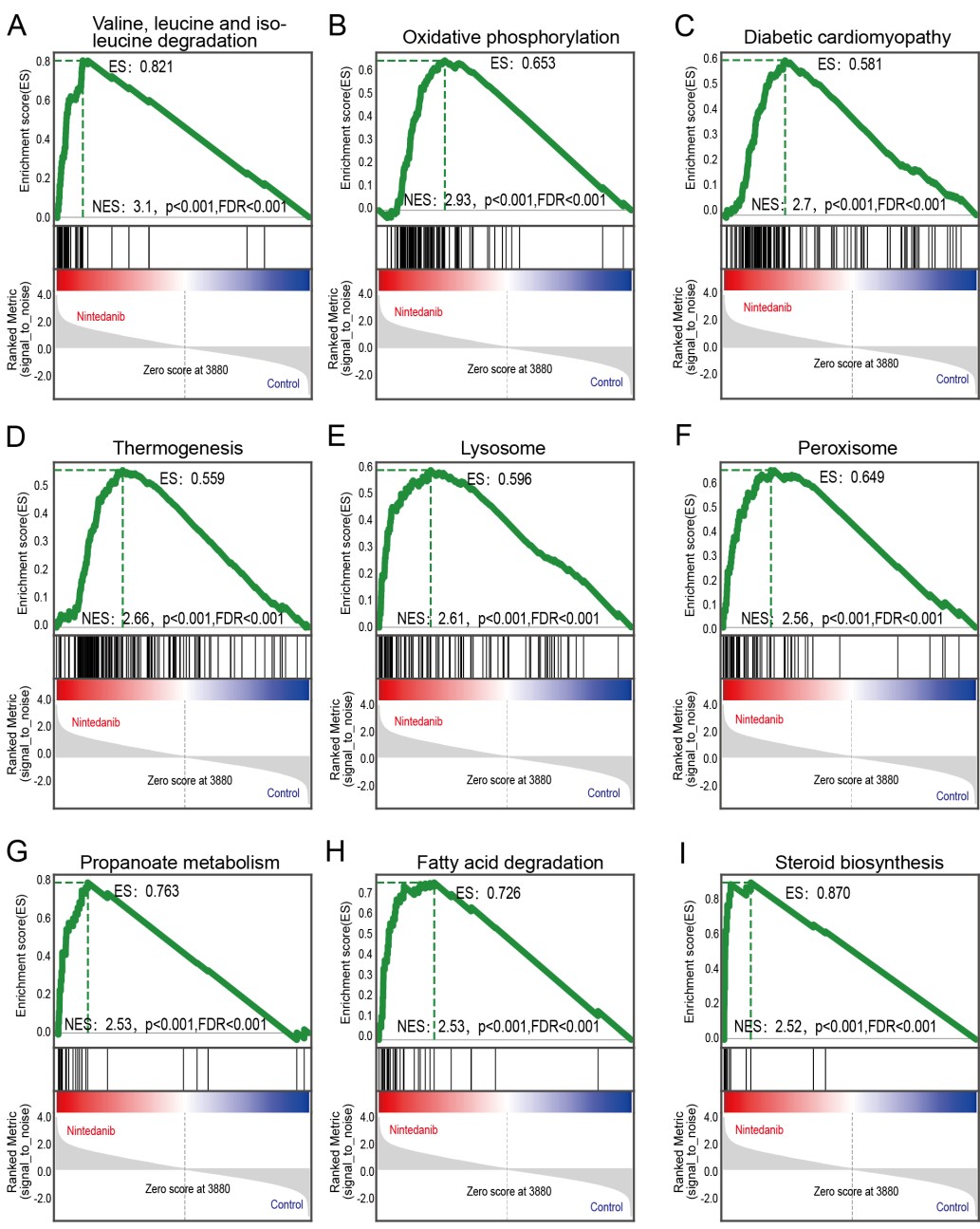

**Figure 5  GSEA pathways enriched in samples with nintedanib treatment.** GSEA pathways enriched in samples after nintedanib treatment. The GSEA results reveal that the terms (A) "Valine, leucine, and isoleucine degradation", (B) "Oxidative phosphorylation", (C) "Diabetic cardiomyopathy", (D) "Thermogenesis", (E) "Lysosome", (F) "Peroxisome", (G) "Propanoate metabolism", (H) "Fatty acid degradation", and (I) "Steroid biosynthesis" were differentially enriched in gastric cancer (GC) samples after nintedanib treatment.

## DISCUSSION

Nintedanib has anti-tumor effects (*Awasthi et al., 2018*; *Corral et al., 2019*; *Won et al., 2019*). It reduces tumor cell proliferation and tumor vascular system growth, elevates

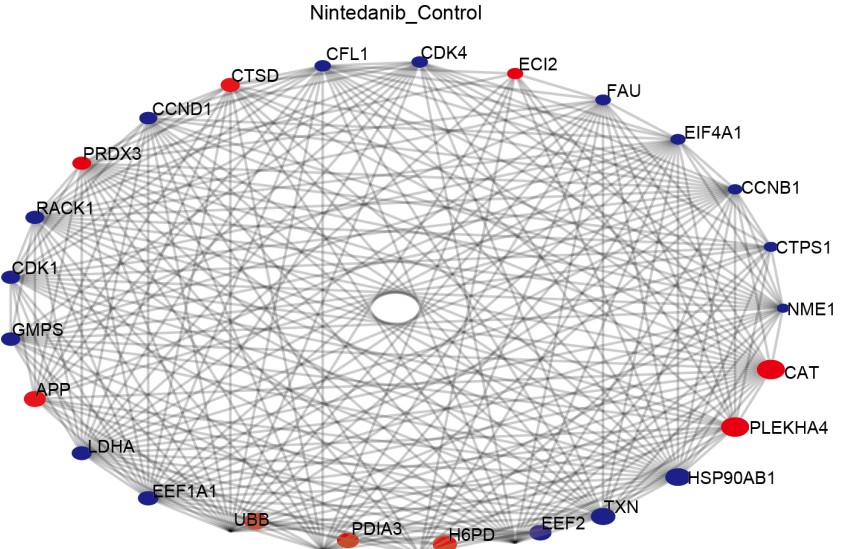

**Figure 6** **Protein–protein interaction network of differentially expressed proteins.** Diagram of protein–protein interaction (PPI) network showing proteins and pathways perturbed by nintedanib treatment. Red circles, upregulated proteins; blue circles, down-regulated proteins. The size of the circle corresponds to the level of connectivity, with larger circles indicating higher connectivity.

**Table 1 Key driver analysis of hub proteins sequencing.**

| Gene name | P | FDR |
| --- | --- | --- |
| EHHADH | 7.53E−07 | 0.000113709 |
| IDH1 | 4.19E−05 | 0.003160212 |
| ACOX1 | 9.59E−05 | 0.004801732 |
| ACAA1 | 0.000127198 | 0.004801732 |
| ACOX3 | 0.000180787 | 0.005459768 |
| ACOX2 | 0.000348173 | 0.007762413 |
| CAT | 0.000359847 | 0.007762413 |
| FUCA2 | 0.000608599 | 0.011487309 |
| SCP2 | 0.000750173 | 0.012586233 |
| PEX2 | 0.001848209 | 0.027907954 |
| TYMS | 0.002383332 | 0.030256861 |
| PEX14 | 0.002571565 | 0.030256861 |
| EIF3B | 0.002665382 | 0.030256861 |
| HSD17B4 | 0.002931307 | 0.030256861 |

tumor cell death, and promotes apoptosis of gastric adenocarcinoma (*Awasthi et al., 2023*). Nintedanib also induces autophagy-dependent cell death in gastric cancer cells by inhibiting STAT3/Beclin1 cascade signaling (*Zhu et al., 2023*). The action mechanisms of antitumor drugs in tumor cells is often complex and not limited to a single death form or signaling pathway (*Vitale et al., 2023*). To date, the published research has largely explored a single action mechanism, but this does not satisfy the current demands. In the present study, we

aimed to evaluate the proteomics phenotype of nintedanib action on gastric cancer cells, and to construct a protein interaction network to study mechanisms of nintedanib action in gastric cancer cells.

Nintedanib is used as an anti-tumor angiogenesis agent for clinical and preclinical studies of various tumors (*Awasthi et al., 2018*; *Hilberg et al., 2008*). In this study, we confirmed that nintedanib directly inhibited gastric cancer cell proliferation, invasion, and migration. The 3D sphere cell culture system results made the findings more reliable. The 3D culture method of tumor cells simulates the growth of tumor cells *in vivo*. Thus, the *in vivo* biological characteristics of tumor cells are more faithfully reproduced compared with the traditional monolayer cell culture (*Ishiguro et al., 2017*). Together with previous research, our results provide explicit evidence that nintedanib has great potential as a single drug in the direct inhibition of tumor cells. To better understand the inhibitory mechanism of nintedanib in gastric cancer cells, additional experiments were performed using quantitative proteome analysis.

TMT-labeled quantitative proteomic analysis was used to identify changes in the proteome in gastric cancer cells after nintedanib treatment for 48 h. Around 845 differentially expressed proteins were identified, 526 up-regulated proteins and 319 down-regulated proteins. GO enrichment analysis revealed that the differentially expressed proteins were significantly enriched in cholesterol metabolic processes and fatty acid oxidation pathways. These differentially expressed protein enrichment results suggest that nintedanib may be involved in growth inhibition of gastric cancer cells by altering specific tumor metabolism. Metabolic tumor reprogramming is one of the malignant tumor markers. Indeed, nintedanib is intricately connected with cellular metabolic processes. Research demonstrates that nintedanib inhibited UGT1A1, a vital metabolic enzyme (*Korprasertthaworn et al., 2019*). Although our enrichment results share no direct correlation with results from the published literature describing nintedanib action mechanisms, these newly identified functional pathways could be promising for future tumor treatment. Cholesterol is an essential molecular component of cells. Abnormal cholesterol anabolism is involved in the progression of various cancers (*Chen et al., 2023*; *Gabitova-Cornell et al., 2020*; *Nelson, Chang & McDonnell, 2014*). Drugs targeting the cholesterol synthesis pathway are already used for tumor therapy (*Liu et al., 2018*). Moreover, abnormal lipid metabolism is an essential metabolic feature that distinguishes tumor cells from normal cells. A high fatty acid oxidation rate is known to provide energy for cancer cell growth and to promote tumor progression, including in gastric cancer (*Carracedo, Cantley & Pandolfi, 2013*; *Chen et al., 2020*; *Wang et al., 2018*).

The differentially expressed proteins affecting biological processes and molecular functions are complex and abundant. Therefore, there is a need for pathway analysis to identify the specific pathways *via* which nintedanib alters molecular functions and biological processes. This study identified overlapping signaling pathways in KEGG and GSEA analyses, and we can infer the possible action mechanism of nintedanib from these pathways. Lysosomes are linked with cell interactions in the tumor microenvironment, and with the proliferation and invasion of tumor cells. They are also associated with tumor

resistance, tumor-associated macrophage polarization, and tumor-associated fibroblasts-mediated downstream mechanisms (*Tan & Finkel, 2022*). The targeting of lysosomes has great potential for the treatment of gastric cancer (*Wang et al., 2023*). *Englinger et al. (2017)* identified the critical features of the lysosomal compartment in tumors, including how the number and size of lysosomes affects the sensitivity of tumor cells to nintedanib. Peroxisomes are ubiquitous in eukaryotic cells, producing many free radicals during oxidative stress. These changes in free radicals are closely related to tumor activity. Studies found that nintedanib improves oxidative stress by down-regulating the PI3K/ Akt/ mTOR signaling pathway, although only in the pulmonary fibrosis model (*Pan et al., 2023*).

The fatty acid degradation pathway is also known to play a vital role in proliferation, metastasis, and signal transduction in tumor cells (*Lee et al., 2020*). Regulation of fatty acid oxidation in gastric cancer is known to inhibit TGF-$\beta$-induced tumor metastasis. Targeting the fatty acid oxidation metabolic pathway could be a new strategy to inhibit gastric cancer metastasis (*Li et al., 2023*). Ferroptosis is another crucial biological process closely associated with fatty acid metabolism, and this metabolic pathway also appears in the KEGG analysis. *Lee et al. (2020)* believe that the fatty acid biosynthesis pathway is decisive in the sensitivity of ferroptosis in gastric cancer. Although there is no direct evidence that nintedanib regulates ferroptosis *in vivo*, ferroptosis is an ROS-dependent form of cell death, and studies have revealed that nintedanib effectively removes ROS (*Li, Chen & Shi, 2022*). The reprogramming of branched-chain amino acid metabolism changes the level of some primary metabolites in tumor cells, including nutrients and ROS required for metabolism. Moreover, these changes continue to stimulate downstream signaling pathways, affecting the survival of tumor cells. Targeting the branched-chain amino acid pathway may be a promising approach in the treatment of cancer (*Peng, Wang & Luo, 2020*). The steroid biosynthesis pathway is also vital in regulating tumors and their microenvironments. Studies indicate that T cell-mediated steroid production in the tumor microenvironment enhances tumor growth by inhibiting anti-tumor immunity (*Mahata et al., 2020*). Finally, propionate metabolism disorder is an essential factor promoting tumors, and it may be useful as a potential target for the treatment of metastatic tumors (*Gomes et al., 2022*).

KDA determines the network regulation of the disease process and its critical nodes based on the topology of biological networks (*Ding et al., 2021*). Nintedanib may be a tumor suppressor in gastric cancer *via* these critical proteins. Isocitrate dehydrogenase (IDH1) expression has been shown to be essential for the survival of gastric cancer cells. This key metabolic enzyme is regulated by HNF4$\alpha$ and is known to play an important role in energy production and in protecting cells from ROS. Drugs targeting wild-type IDH1 could be clinically helpful in some gastric cancers (*Xu et al., 2020*). Alpha-L-fucosidase 2 (FUCA2) is an essential catalytic factor in the fucosylation of gastric cancer cells, and FUCA2 fucosidase activity is associated with tumor formation, metastasis inhibition, and multi-drug resistance. Immunohistochemistry has revealed that FUCA2 expression is significantly increased in gastric cancer tissues, and FUCA2 expression levels are correlated with surgical stage and advanced histological grade (*Leal Quirino et al., 2022*). Eukaryotic Translation Initiation Factor 3 Subunit B (EIF3B) is a critical factor in the protein synthesis

pathway. EIF3B promotes tumor migration and metastasis by activating the PI3K/ AKT/ mTOR pathway during the development of gastric cancer (*Wang et al., 2019*). Several of the differentially expressed proteins have been demonstrated to affect other cancer types, although an association with gastric cancer has not previously been established. Acetyl-CoA Acyltransferase 1 (ACAA1) is the last metabolic enzyme in the long-chain fatty acid metabolic pathway. Although a role for ACAA1 in gastric cancer has not been clarified, researchers observed that ACAA1 down-regulation inhibited the proliferation of triple-negative breast cancer cells and enhanced their response to CDK4/6 inhibitors (*Peng et al., 2023*). Catalase (CAT) is an essential antioxidant enzyme in most organisms. CAT inhibitors are known to increase ROS production in cancer cells and to induce ferroptosis (*Cao et al., 2023*). Sterol Carrier Protein 2 (SCP2) is a lipid transporter that plays a vital role in cellular cholesterol synthesis and metabolism (*Schroeder et al., 2010*). SCP2 expression changes may cause intracellular metabolic abnormalities. SCP2-specific inhibitors are known to induce autophagy in tumor cells by inhibiting the AKT1-mTOR signaling pathway, thereby suppressing tumor cell proliferation (*Liu et al., 2014*). Hydroxysteroid 17-beta dehydrogenase 4 (HSD17B4) protein (D-bifunctional protein) is located in peroxisomes, where it plays a central role in fatty acid $\beta$-oxidation (*Breitling et al., 2001*). There is evidence that HSD17B4 is associated with a variety of cancers, including liver cancer, ovarian cancer, and prostate cancer, making it a potential therapeutic target (*Ko et al., 2018*; *Lu et al., 2019*; *Nagayoshi et al., 2005*). Several other proteins are essential in the pathway that nintedanib may act on. Enoyl-coenzyme A, Hydratase/3-hydroxyacyl coenzyme A dehydrogenase (EHHADH), Acyl-CoA oxidase 1 (ACOX1), Acyl-CoA oxidase 2 (ACOX2), and Acyl-CoA oxidase 3 (ACOX3) are all important enzymes in the classical peroxisome fatty acid $\beta$-oxidation pathway, where they play a role in pathway maintenance (*Houten et al., 2012*; *Ranea-Robles et al., 2021*; *Zhang et al., 2023*). Peroxisomal Biogenesis Factor 2 (PEX2) and Peroxisomal Biogenesis Factor 14 (PEX14) are proteins involved in peroxisome biogenesis. Inhibiting their function affects the normal operation of the peroxisome (*Feng et al., 2022*; *Okumoto et al., 2020*).

Finally, we acknowledge our study has certain limitations. Most importantly, additional biochemical experiments are required to help validate the pathway changes and the key proteins that were identified through our study on the control and nintedanib treatment groups.

## CONCLUSIONS

The present study reconfirmed the toxic effects of nintedanib on gastric cancer cells, providing evidence for nintedanib inhibition of their proliferation, invasion, and metastasis. Furthermore, proteomic analysis demonstrated that tumor metabolism-related pathways play important roles in the inhibition of gastric cancer cells by nintedanib. The affected pathways include: branched-chain amino acid metabolism, steroid biosynthesis, propionate metabolism, fatty acid metabolism, lysosomes, peroxisomes, and ferroptosis. In addition, several differentially expressed proteins, including EHHADH, IDH1, ACOX1, ACAA1, and ACOX3, could be important targets for its action. Nintedanib may represent a promising

new approach to gastric cancer treatment. However, the specific mechanism must be verified and further clinical trials are needed.

### Funding

This work was supported by the National Health Commission Key Laboratory of Gastrointestinal Tumor Diagnosis and Treatment 2022 Master/Postdoctoral Fund Program (NHCDP2022015); the Key Laboratory of Diagnosis and Therapy of Gastrointestinal Tumor of National Health Commission (2019PT320005); and the Key Talent Project of the Organization Department of Gansu provincial Party Committee (2020RCXM076). The funders had no role in study design, data collection and analysis, decision to publish, or preparation of the manuscript.

### Grant Disclosures

The following grant information was disclosed by the authors:
National Health Commission Key Laboratory of Gastrointestinal Tumor Diagnosis and Treatment 2022 Master/Postdoctoral Fund Program: NHCDP2022015.
Key Laboratory of Diagnosis and Therapy of Gastrointestinal Tumor of National Health Commission: 2019PT320005.
Key Talent Project of the Organization Department of Gansu provincial Party Committee: 2020RCXM076.

### Competing Interests

The authors declare there are no competing interests.

### Author Contributions

- Xiaohua Dong conceived and designed the experiments, performed the experiments, analyzed the data, prepared figures and/or tables, authored or reviewed drafts of the article, and approved the final draft.
- Liuli Wang performed the experiments, authored or reviewed drafts of the article, and approved the final draft.
- Da Wang performed the experiments, authored or reviewed drafts of the article, and approved the final draft.
- Miao Yu analyzed the data, prepared figures and/or tables, and approved the final draft.
- Xiao jun Yang analyzed the data, prepared figures and/or tables, and approved the final draft.
- Hui Cai conceived and designed the experiments, authored or reviewed drafts of the article, and approved the final draft.

### Data Availability

The raw images and cell line experimental raw data are available at figshare:
Dong, Xiaohua; Wang, Liuli; Wang, Da; Yu, Miao; Yang, Xiao-Jun; Cai, Hui (2024).

Proteomic study of nintedanib in gastric cancer cells. figshare. Figure. Available at
https://doi.org/10.6084/m9.figshare.23804079.v1

The proteomic raw data is available at iProX/ProteomeXchange: IPX0006789000,
PXD044051.

https://www.iprox.cn/page/project.html?id=IPX0006789000
https://proteomecentral.proteomexchange.org/cgi/GetDataset?ID=PXD044051

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
