# Peer review of "Proteomic study on nintedanib in gastric cancer cells"

_PeerJ, doi:10.7717/peerj.16771_

## Round 0.1 · original submission · Major Revisions

All three reviewers gave suggestions for modification. Please revise carefully and answer the author's questions.

**Language Note:** The review process has identified that the English language must be improved. PeerJ can provide language editing services - please contact us at [email protected] for pricing (be sure to provide your manuscript number and title). Alternatively, you should make your own arrangements to improve the language quality and provide details in your response letter. – PeerJ Staff

Reviewer 1 ·

Basic reporting

This is a straightforward and well-written study of Dong et al. describing a proteomics dataset for the activity of nintedanib against two gastric cancer cell models. The proteomic experiment is supported by a limited amount of in vitro data demonstrating anti-proliferative and –migratory activity in this setting. The study is, however, not very innovative and any confirmative experiments of the detected pathways etc. are missing.

Major points of concern before publication:
Generally, the discussion of the literature in the introduction and discussion are highly random and should be focused on the literature already available for nintedanib. For example, several other proteomic datasets for nintedanib exist in other setting. Differences and similarities should be discussed. But in the introduction just random examples where proteomic analyses supported drug development are given. This does not make sense and needs focus. Same, in the discussion section concerning the pathways and organelles detected in the data evaluation. Especially, in the literature concerning fibrotic events interactions of nintedanib with fatty acid metabolism have been indicated and interaction with e.g. lysosomes have been described. In summary, the data should be evaluated against the literature.

Experimental design

The study seems well performed and the data analysis well done.

Validity of the findings

The data, though rather limited, appear valid and relevant!

Additional comments

µM is micromolar but µm is micrometer! Please correct in the different panels of figure 1.

Cite this review as

·

Basic reporting

There are no major flaws in basic reporting but please check the manuscript for grammar and spelling errors.

Experimental design

The manuscript titled "Proteomic study of nintedanib in gastric cancer cells" by Dong et al. investigated the potential mechanisms and targets of nintedanib action on gastric cancer cells, shedding light on its promising role in oncology treatment. The authors employed high-throughput proteomic approach and bioinformatic analysis to explore the systemic effects of nintedanib on gastric cancer cells. Overall, this study may help in advancing our understanding of nintedanib's effects on gastric cancer cells. However, there are several critical points that need to be addressed:

Major comments:

1. The authors analyzed the effects of nintedanib on two different gastric cancer cells, AGS and MKN28 but only performed proteomic analysis using AGS cells. Different cell lines may respond differently to treatments or exhibit variations in molecular profiles. The use of a single cell line may not adequately represent the diversity of gastric cancer and might have a higher risk of bias.

2. What is the actual IC50 value of nintedanib on the gastric cancer cells? This value is critical as it quantifies the drug's effectiveness and its concentration needed for inhibitory effects.

Validity of the findings

Major comments:

3. The description of the experimental procedures and data analysis is not sufficiently reported, especially in the context of proteomics analysis. The authors should follow the guidelines for reporting proteomic data - Minimum Information About a Proteomics Experiment (MIAPE). Details about the instrument's parameter settings, acquisition method, search parameters etc should be provided. The study should also provide more detailed information regarding the data analysis.

4. The study identifies a substantial number of differentially expressed proteins upon nintedanib treatment. It is essential to discuss the potential functional significance of these proteins in the context of gastric cancer and their role in the mechanisms of action of nintedanib.

5. While the study mentions enriched pathways and key drivers, a more detailed interpretation of these findings is needed. How do these pathways relate to the anti-cancer effects of nintedanib? What are the potential clinical implications of targeting these pathways?

Additional comments

Minor comments:

1. Line 77: What is the manufacturer of CCK-8?

2. Line 79: What is "the appropriate number of cells"?

3. Line 84: What is "enzyme marker FC"?

4. In Figure 1A, does the y-axis represent absorbance or cell viability?

·

Basic reporting

The language, references, article structure are appropirate.

Experimental design

Design and exeution of the experiments is carried out systematically.

Validity of the findings

The findings are intersteing and novel

Additional comments

1. Line 110, more details would be useful to provide liquid chromatography conditions like detection wavelength, flow rate, gradient and mobile phase conditions.
2. Line 125, while measuring the mean +/- SD, details to be mentioned here like how many replicates and how those replicates were designed though it was mentioned in line 147.
3. Seems a typo in line 144 “and 10 µm3”.
4. Type in Figure 1A, it was mentioned as KKN-28 instead of MKN28.
5. Labels in Figure 1D, the labels in the statistical graph are not clearly visible.
6. Its very difficult to read the molecule names in figure 3A especially the green ones, can other ways of representation be explored?
7. The lines from 203 to 210 seems more belong to the introduction part rather than here, as the actual discussion related to the results starts from line 210.
8. The results obtained from Key Drive Analysis are promising, there should be more discussion extending 267- 283 lines
9. Figure 5 caption is repeated.
10. In the discussion for upregulated and down regulated proteins, few more sentences can be added with details (Lines 229-240)
11. Couple of sentences are required for future prospects and limitations (lines 282-283)
12. Figure 6 representation is appreciated

---

## Round 0.2 · Minor Revisions

The author is requested to make further revisions.

·

Basic reporting

This manuscript is significantly improved that the first version which I had reviewed before. However, there are still some minor language glitches. For example, in material and methods, the description on proteomics and mass-spec methods are a bit weird (All the MS/MS spectra were collected using high-energy collision "cracking" in da-ta-dependent positive ion mode using a normalized collision energy of 32). Please edit and correct these errors.

Experimental design

It is ok although I would prefer more cell lines to be analyzed.

Validity of the findings

No comment

Additional comments

The authors have addressed all my previous concerns. I have no more comments; except that the authors should go for another round of language check with preferably fluent English-speaker scientists to make sure that the language is understandable by international scientists working in the same field.

·

Basic reporting

All the data presented is appropriate. All the comments are addressed appropriately.

Experimental design

All the comments are addressed appropriately.

Validity of the findings

All the comments are addressed appropriately.

Additional comments

All the comments are addressed appropriately. I have no further comments.

---

## Round 0.3 · accepted · Accept

Although one reviewer has not responded to the review invitation, two experts have agreed to accept the manuscript, which meets the requirements. I also reviewed the manuscript and found no significant risk to publication, so I agreed to publish this article.

·

Basic reporting

The authors have addressed my previous comments.

Experimental design

No further comments

Validity of the findings

No further comments

Additional comments

No further comments